# Numerical Study of Nonadiabatic Wall Effects on Aerodynamic Characteristics of CHN-T1 Standard Model

**Xiaobing Wang** [1], **Junqiang Wu** [2], **Jianzhong Chen** [1], **Yuping Li** [1], **Zhongliang Zhao** [1], **Guangyuan Liu** [1], **Yang Tao** [1,*] **and Neng Xiong** [1,*]

1   High Speed Aerodynamics Institute, China Aerodynamics Research and Development Center, Mianyang 621000, China
2   China Aerodynamics Research and Development Center, Mianyang 621000, China
*   Correspondence: taoyang@cardc.cn (Y.T.); xiongneng@cardc.cn (N.X.); Tel.: +86-136-2808-1890 (Y.T.)

**Abstract:** Cryogenic wind tunnels provide the for possibility aerodynamic tests to take place over high Reynolds numbers by operating at a low gas temperature to meet the real flight simulation requirements, especially for state-of-the-art large transport aircrafts. However, undesirable temperature gradients between the test model and the surroundings will be caused by the thermal non-equilibrium, changing the boundary layer characteristics and resulting in test errors. To study the nonadiabatic wall effects on the aerodynamic characteristics of the model in cryogenic wind tunnels, a numerical study was carried out for the CHN-T1 standard model under different wall temperature gradients. A code with a finite volume method and $\gamma$-$Re_{\theta t}$ transition model were used. The analysis concluded that the change in wall temperature significantly affects the surface pressure distribution, transition position and skin-friction coefficient of the model, thus varying the lift and drag coefficients of the aircraft. The influences on the flow characteristics of both laminar and turbulent boundary layers by the wall temperature gradient were also investigated.

**Keywords:** cryogenic wind tunnel; high Reynolds number; CHN-T1; temperature; numerical simulation





## 1. Introduction

State-of-the-art large transport aircrafts usually adopt supercritical wings. Flow patterns exist that are directly influenced by Reynolds number, including the boundary layer development and transition, and shock/boundary layer interference. The large scale of the aircrafts makes the Reynolds number difficult to achieve in conventional wind tunnels, so artificial transition techniques and some other correction methods should be used during the application of wind tunnel test results in real flights.

The cryogenic wind tunnel provides the aerodynamic tests with the ability to be conducted over high Reynolds numbers by injecting liquid nitrogen into the circuit to lower the gas temperature to meet the real flight simulation requirements and accurately predict the aerodynamic characteristics. Nevertheless, the low gas temperature brings another problem: the temperature gradient and heat transfer between the test model and the incoming flow in the wind tunnel will change the boundary layer characteristics to some non-negligible extent, as well as the shock wave position and the flow separation. Thus, the accurate prediction of the aerodynamic characteristics may be affected.

At present, there are some cryogenic wind tunnels, including two large-scale production transonic wind tunnels, the European Transonic Wind Tunnel (ETW) and the National Transonic Facility (NTF). To build efficient operation norm and scientific test technology for cryogenic wind tunnels, numerous studies have been carried out, focusing on the high Reynolds number flow mechanism, aerodynamic characteristics, and thermodynamic similarity under low-temperature conditions.

Early research mainly focused on the laminar boundary layer. Johnson [1] analyzed the time-varying effect of nonadiabatic wall conditions on boundary layer properties for

a two-dimensional wing section and a revolution typical of a fuselage, and indicated how the model skin-friction drag and boundary layer displacement thickness change when the total temperature of the wind tunnel decreases under different total pressure. Macha [2] studied the influence of nonadiabatic wall conditions on skin-friction drag and boundary layer displacement thickness using a model of built-in heat source in a cryogenic wind tunnel, and concluded that local surface temperature deviations of the order of tens of degrees Kelvin above adiabatic are necessary before the validity of the wind tunnel simulation is affected. Mabey [3] summarized the influence of heat conduction on the static aerodynamic characteristics of the model, quoting the conclusion of VanDriest [4]: wall cooling will increase the density, decreasing the thickness of the boundary layer and making the velocity distribution more "full", causing an increase in the velocity gradient at the wall. This, together with a small decrease in the viscosity coefficient, leads to an increase in the skin-friction drag. However, the greatly reduced boundary layer thickness reduces the form drag to a greater extent. Therefore, the total drag will be decreased, as observed in steady experiments [5,6]. Feiler [7] analyzed how the skin-friction drag of the laminar boundary layer changes with the wall temperature, especially in the presence of non-zero pressure gradients. It was pointed out that the heat transfer effect of varying viscosity is dependent on other influences, while the effect of density changes is enhanced by adverse pressure gradients and diminished by favorable pressure gradients. For the turbulent boundary layer, both Maybe and Feiler believed that the specified findings are also qualitatively valid.

To determine the heat transfer effect on the stability and transition of a laminar boundary layer, extensive research has also been carried out. Lees [8] found that wall cooling can stabilize the two-dimensional disturbances in the plate boundary layer. Boehman [9] found that wall cooling can stabilize two-dimensional and three-dimensional disturbances in transonic flow. Potter [10] summarized the tests on wall cooling at hypersonic speed and found that, in most cases, the wall transition Reynolds number increases with the increase in cooling degree. Dong [11] calculated the boundary layer transition characteristics under different plate boundary layer temperature gradients. The results show that higher temperature gradients can delay the transition, but their effect on transition is weaker than other factors, such as turbulent intensity, freestream velocity and pressure gradient. Wu [12] studied the influence of wall temperature on the plate laminar boundary layer and turbulent boundary layer by applying temperature control to plate boundary layer flow with a zero-pressure gradient. The results show that wall heating increases the viscosity of the boundary layer in the laminar region, reduces the Reynolds shear stress and turbulent kinetic energy, stabilizes the flow and delays the transition. However, for the turbulent region, wall heating decreases the velocity gradient and viscous shear stress in the viscous sub-layer, resulting in a reduction in the skin-friction drag. Costantini [13] analyzed the effects of pressure gradient and nonadiabatic wall effect on boundary layer transition through experimental methods, and indicated that a stronger flow acceleration and lower wall temperature ratios lead to an increase in the transition Reynolds number, and the relative variation in the transition Reynolds number as a function of the wall temperature ratio was found to be reasonably approximated by the minus-four power law of the wall temperature ratio.

A considerable number of studies on the effects of wall temperature gradient and heat transfer on aerodynamic characteristics have been conducted, mainly concentrating on boundary layer flow profiles and laminar flow stability and transition. The present study is focusing on the nonadiabatic wall's effect on the steady aerodynamic coefficients of a transport aircraft model in a cryogenic wind tunnel. The Langtry–Menter $\gamma$-$Re_{\theta t}$ transition model was adopted to numerically simulate the flow characteristics of a CHN-T1 standard model under different wall-temperature ratios. The nonadiabatic wall effects on lift and drag coefficients, together with the surface and boundary layer flow characteristics, were investigated.

## 2. Computation Framework

An in-house CFD code was used in the present work, which is a 3D, cell-centered Reynolds-averaged Navier–Stokes (RANS) solver, based on a structured grid, and implemented with essential turbulent and transition models. The multigrid and grid sequencing scheme were adopted to accelerate convergence. The parallel mode was also available to improve iteration speed. Some basic control equations and schemes are illustrated in the present section.

### 2.1. Governing Equations and Discrete Scheme

The three-dimensional Reynolds-averaged Navier–Stokes equations in a curvilinear coordinate system can be described as:

$$\frac{\partial \hat{Q}}{\partial t} + \frac{\partial \left( \hat{F} - \hat{F}_v \right)}{\partial \xi} + \frac{\partial \left( \hat{G} - \hat{G}_v \right)}{\partial \eta} + \frac{\partial \left( \hat{H} - \hat{H}_v \right)}{\partial \zeta} = 0 \tag{1}$$

where $\hat{Q}$ is the vector of conservative variables, $\hat{F}$, $\hat{G}$, $\hat{H}$ are inviscid flux vectors in $\xi$, $\eta$, $\zeta$ directions, and $\hat{F}_v$, $\hat{G}_v$, $\hat{H}_v$ are the viscous flux vectors.

The spatial discretization adopted the finite volume method based on a multi-block structural grid, in which the inviscid flux is discretized by Roe's flux difference scheme, and the conservative variables at the interface are interpolated by a third-order upwind scheme for conservation laws (MUSCL). For the discontinuous problems, the Venkatakrishnan limiter was used to suppress numerical oscillation. The viscous flux was solved by the central difference scheme. The solution that converges with time was obtained by the implicit scheme of Lower-Upper Symmetric Gauss–Seidel (LU-SGS). In addition to the freestream boundary condition used for the far-field, no-slip wall conditions, adiabatic or isothermal, were adopted for the model surface.

### 2.2. Turbulent and Transition Model

The $k$-$\omega$ shear stress transport (SST) turbulence model was used, which was first proposed by Menter in 1994 [14,15], combining the advantages of the $k$-$\varepsilon$ model away from the wall and $k$-$\omega$ model near the wall. The SST model is widely used in turbulent flow simulations due to its accuracy and efficiency.

The governing equation of the SST model is as follows:

$$\frac{\partial(\rho k)}{\partial t} + \frac{\partial(\rho U_i k)}{\partial x_i} = \widetilde{P}_k - \rho \beta^* k \omega + \frac{\partial}{\partial x_i}\left[(\mu + \sigma_k \mu_t)\frac{\partial k}{\partial x_i}\right] \tag{2}$$

$$\frac{\partial(\rho \omega)}{\partial t} + \frac{\partial(\rho U_i \omega)}{\partial x_i} = \frac{\gamma}{\mu_t} P_k - \rho \beta \omega^2 + \frac{\partial}{\partial x_i}\left[(\mu + \sigma_\omega \mu_t)\frac{\partial \omega}{\partial x_i}\right] + 2(1 - F_1)\rho \sigma_{\omega 2}\frac{1}{\omega}\frac{\partial k}{\partial x_i}\frac{\partial \omega}{\partial x_i} \tag{3}$$

$$\mu_t = \rho a_1 \frac{k}{max(a_1 \omega, F_2 S)} \tag{4}$$

where $k$ is the turbulent kinetic energy, $\omega$ is the dissipation rate, $\mu$ is the laminar viscosity coefficient, $\mu_t$ is the turbulent viscosity coefficient, $\rho$ is density, $U$ is velocity, $x$ is coordinates, the production term $P_k$ is defined as:

$$P_k = \left[\mu_t\left(\frac{\partial U_i}{\partial x_j} + \frac{\partial U_j}{\partial x_i} + \frac{2}{3}\frac{\partial U_k}{\partial x_k}\delta_{ij}\right) - \frac{2}{3}\rho k \delta_{ij}\right]\frac{\partial U_i}{\partial x_j} \tag{5}$$

and the term was limited as:

$$\widetilde{P}_k = min(P_k, 10\rho\beta^* k\omega) \tag{6}$$

The two blending functions used to adjust the closure constants in different flow regions are defined as follows:

$$F_1 = \tanh\left\{ \left[ \min\left( \max\left( \frac{\sqrt{k}}{\beta^* \omega d}, \frac{500\nu}{d^2\omega} \right), \frac{4\rho\sigma_{\omega_2}k}{CD_{k\omega}d^2} \right) \right]^4 \right\} \tag{7}$$

$$F_2 = \tanh\left\{ \left[ \max\left( \frac{2\sqrt{k}}{\beta^* \omega d}, \frac{500\nu}{d^2\omega} \right) \right]^2 \right\} \tag{8}$$

where

$$CD_{k\omega} = \max\left( 2\rho\sigma_{\omega_2}\frac{1}{\omega}\nabla k \cdot \nabla\omega, 10^{-10} \right) \tag{9}$$

Model constants $\beta$, $\gamma$, $\sigma_k$, $\sigma_\omega$ are blended in the form of

$$\phi = F_1\phi_1 + (1 - F_1)\phi_2 \tag{10}$$

where $\beta_1 = 0.075$, $\beta_2 = 0.0828$, $\gamma_1 = \frac{\beta_1}{\beta^*} - \sigma_{\omega_1}\frac{\kappa^2}{\sqrt{\beta^*}}$, $\gamma_2 = \frac{\beta_2}{\beta^*} - \sigma_{\omega_2}\frac{\kappa^2}{\sqrt{\beta^*}}$, $\sigma_{k_1} = 0.85$, $\sigma_{k_2} = 1.0$, $\sigma_{\omega_1} = 0.5$, $\sigma_{\omega_2} = 0.856$. For other constants, $\beta^* = 0.09$, $\kappa = 0.41$, $a_1 = 0.31$.

The $\gamma$-$Re_{\theta t}$ transition model was adopted in the present study, which was first proposed by Menter and Langtry in 2004 [16]. The model is composed of two transport equations: intermittent factor transport equation and momentum thickness Reynolds number transport equation. The model solves the intermittent factor and determines the transition behaviour.

The intermittent factor transport equation is:

$$\frac{\partial(\rho\gamma)}{\partial t} + \frac{\partial(\rho U_j \gamma)}{\partial x_j} = P_\gamma - E_\gamma + \frac{\partial}{\partial x_j}\left[ \left( \mu + \frac{\mu_t}{\sigma_\gamma} \right)\frac{\partial\gamma}{\partial x_j} \right] \tag{11}$$

where $\gamma$ is the intermittent factor; the production and destruction term are defined as:

$$P_{\gamma 1} = F_{length}c_{a1}\rho S[\gamma F_{onset}]^{0.5}(1 - \gamma) \tag{12}$$

$$E_\gamma = c_{a2}\rho\Omega F_{turb}(c_{e2}\gamma - 1) \tag{13}$$

where $S$ is the strain rate; $F_{length}$ is the length of transition zone defined using the transition momentum thickness Reynolds number $\overline{Re}_{\theta t}$:

$$F_{length} = \min\left[ 0.5 + \exp(7.168 - 0.01173\overline{Re}_{\theta t}), 300 \right] \tag{14}$$

Control functions $F_{onset}$ and $F_{turb}$ are defined as:

$$F_{onset} = \max\left\{ \min\left[ \max\left( \left( \frac{Re_\nu}{C_{onset1}Re_{\theta c}} \right)^4, \frac{Re_\nu}{C_{onset1}Re_{\theta c}} \right), 2 \right] - \max\left[ 1 - \left( \frac{Re_t}{2.5} \right)^3, 0 \right], 0 \right\} \tag{15}$$

$$F_{turb} = \exp\left[ -\left( \frac{Re_t}{4} \right)^4 \right] \tag{16}$$

where $\Omega$ is the vorticity; $c_{a1} = 2$, $c_{a2} = 0.06$, $c_{e2} = 50$, $\sigma_\gamma = 1$, $C_{onset1} = 2.193$ are model constants.

The momentum thickness Reynolds number transport equation is:

$$\frac{\partial(\rho\overline{Re}_{\theta t})}{\partial t} + \frac{\partial(\rho U_j\overline{Re}_{\theta t})}{\partial x_j} = P_{\theta t} + \frac{\partial}{\partial x_j}\left[ \sigma_{\theta t}(\mu + \mu_t)\frac{\partial(\overline{Re}_{\theta t})}{\partial x_j} \right] \tag{17}$$

The production term $P_{\theta t}$ is defined as:

$$P_{\theta t} = C_{\theta t} \frac{\rho}{t} \left( Re_{\theta t} - \overline{Re}_{\theta t} \right) \left( 1 - F_{\theta t} \right) \tag{18}$$

$$F_{\theta t} = min\left\{ max\left[ F_{wake}exp\left( -\left( \frac{\rho U^2}{375 W \mu \overline{Re}_{\theta t}} \right)^4 \right), 1 - \left( \frac{C_{e2}\gamma - 1}{C_{e2} - 1} \right)^2 \right], 1 \right\} \tag{19}$$

$$F_{wake} = exp\left[ -\left( \frac{\rho \omega d^2}{10^5 \mu} \right)^2 \right] \tag{20}$$

where $d$ is the distance to the nearest wall boundary, $C_{\theta t} = 0.03$, $\sigma_{\theta t} = 2$ are model constants.

## 3. Model and Grids

### 3.1. Model

The CHN-T1 model was used in the present study, which is a standard model for a civil, single-aisle passenger aircraft developed by the China Aerodynamics Research and Development Center (CARDC) [17] when performing a credibility validation of both CFD simulation and a wind tunnel test. The plane has a similar layout to B737, A320 and C919, which comprise supercritical wing, single-aisle fuselage, horizontal and vertical tails, nacelle and pylon. The body + wing + horizonal tail + vertical tail configuration (BWHV) was adopted for simplicity in the present work. An aerodynamic test was conducted in the 2.4 m transonic wind tunnel of China Aerodynamics Research and Development Center [18] using the tail sting support scheme. The blockage of the model in the test section is 0.97%. Cylindrical transition trips with different heights were adopted for the fixed transition simulation. For the wings, and the horizonal and vertical tails, the 0.1 mm trips were pasted at 7% local chord length from the leading edges, and for the fuselage, the 0.18 mm trips were pasted 25 mm from the top of the nose. Figure 1 shows the complete configuration and a photo of the wind tunnel test. A 1:19.23 scaled model was used in the simulation, of which the reference area or the wing area $S = 0.2578$ m$^2$, the wingspan $b = 1.5482$ m, the aerodynamic mean chord length $c_A = 0.1937$ m, and the fuselage length was $L = 1.5744$ m.

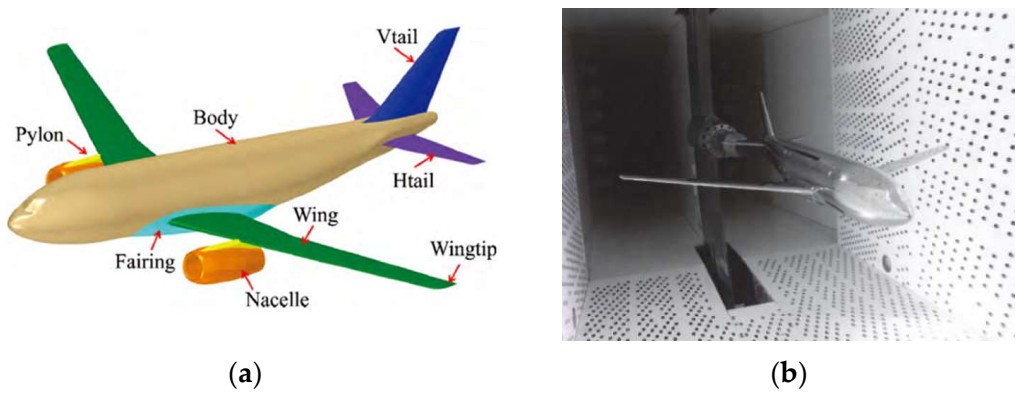

(**a**)           (**b**)

**Figure 1.** The complete configuration of CHN-T1 (**a**) and the model tested in 2.4 m wind tunnel (**b**). Reprinted with permission from Ref. [18], 2019, Li Q.

### 3.2. Grid Independence Study

Multi-block structural grids were applied with an H-type distribution, discretizing a symmetric half of the model. Firstly, the impact of the grid size on the simulation results should be studied before choosing an appropriate set of grids. Three sets of grids with different densities were adopted: the coarse grid (with 6 million cells), the medium grid (with 12 million cells), and the fine grid (with 24 million cells). The first grid spacing near the wall for each set of grids was set to $1.2 \times 10^{-4}$ mm to make $y^+ < 1.0$, even for the case when $Re = 5.0 \times 10^7$. Some other geometric parameters are listed in Table 1, where

*OceNo.* represents overall cell numbers, *LCeNo.* represents cell numbers along the leading edge, *LSC.min* and *LSC.max* represent the minimum and maximum of the local spacing in percentage along the chord direction where the fuselage and wing intersect, and *LSS.min* and *LSS.max* represent the minimum and maximum of the local spacing in percentage along the wingspan direction. The surface grid distributions are shown in Figure 2. The grid independence study took place under the flow conditions of $M = 0.78$, $Re = 3.3 \times 10^6$ and $\alpha = 2°$; the model surface was adiabatic and flow transition was numerically predicted.

**Table 1.** Important parameters for different sets of grids.

| Grid | *OCeNo.* (in Million) | *LCeNo.* | *LSC.min* | *LSC.max* | *LSS.min* | *LSS.max* |
|---|---|---|---|---|---|---|
| Coarse grid | 6.15 | 11 | 0.36% | 3.10% | 0.06% | 1.66% |
| Medium grid | 12.08 | 17 | 0.23% | 2.07% | 0.04% | 1.15% |
| Fine grid | 23.95 | 23 | 0.16% | 1.21% | 0.03% | 0.83% |

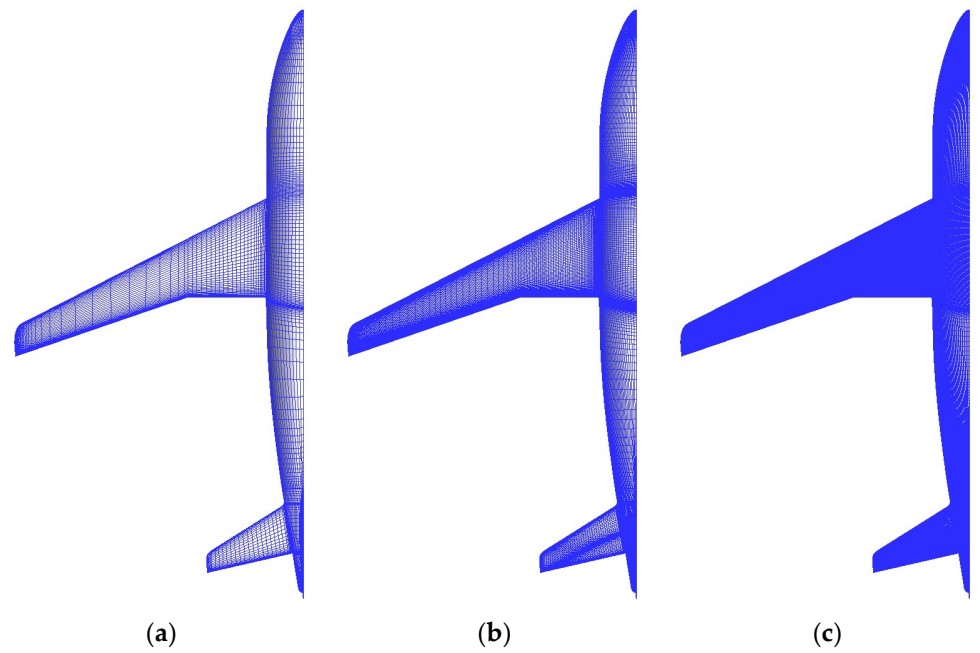

(**a**)           (**b**)           (**c**)

**Figure 2.** Surface grid distribution. (**a**) Coarse grid; (**b**) medium grid; (**c**) fine grid.

Table 2 shows the calculation results and the corresponding experimental results. $C_L$, $C_D$ and $C_m$ are nondimensional results for lift, drag and pitching moment, defined as:

$$C_L = \frac{L}{0.5\rho U^2 S}, C_D = \frac{D}{0.5\rho U^2 S}, C_m = \frac{M}{0.5\rho U^2 S c_A} \tag{21}$$

where $L$, $D$, $M$ represent lift, drag and pitching moment. Figure 3 shows the convergence history of residuals and aerodynamic coefficients, and the grid distribution of the wing surface of the medium grid; using these, the accuracy and efficiency of the simulation are both acceptable. In Table 2, there are only slight differences when comparing CFD results with the medium grid with the experimental results, except for $C_m$ because of the sting interference. Therefore, the medium grid size is suitable for the free transition simulation and was adopted in the present work.

**Table 2.** Influence of grid size on aerodynamic characteristics.

| Scheme | $C_L$ | $C_D$ | $C_m$ |
|---|---|---|---|
| CFD (Coarse Grid) | 0.4202 | 0.0265 | 0.0624 |
| CFD (Medium Grid) | 0.4517 | 0.0243 | 0.0461 |
| CFD (Fine Grid) | 0.4552 | 0.0241 | 0.0425 |
| Experiment | 0.4454 | 0.0247 | 0.0024 |

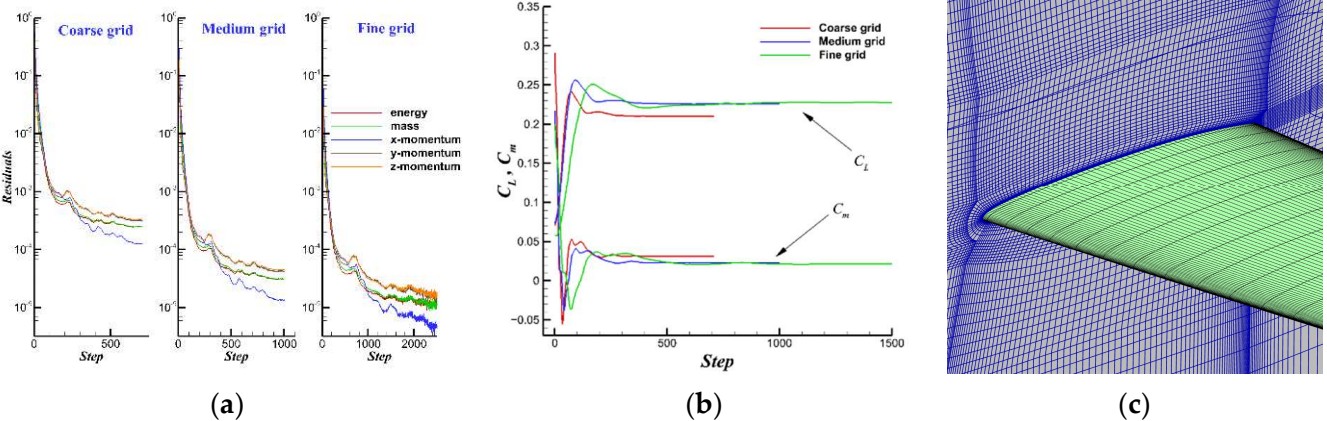

(**a**)      (**b**)      (**c**)

**Figure 3.** Convergence history of (**a**) residuals and (**b**) aerodynamic coefficients; (**c**) grid distribution of medium grid.

## 4. Results and Discussion

### 4.1. Computational Validation with Wind Tunnel Test Results

In order to verify the numerical method, including the SST turbulence model and Langtry–Menter transition model, Case 1 was simulated for the CHN-T1 model. The calculation parameters are shown in Table 3.

**Table 3.** Parameters of Case 1.

| $M$ | $Re$ | $\alpha$ | $T_{inf}$ | Wall Boundary Condition |
|---|---|---|---|---|
| 0.78 | $3.3 \times 10^6$ | $-2°, 0°, 2°, 4°$ | 256.3K | adiabatic |

Figure 4 shows the distribution of the skin-friction coefficient for free transition obtained by CFD, and Figure 5 shows the comparison between the CFD results and the wind tunnel test results.

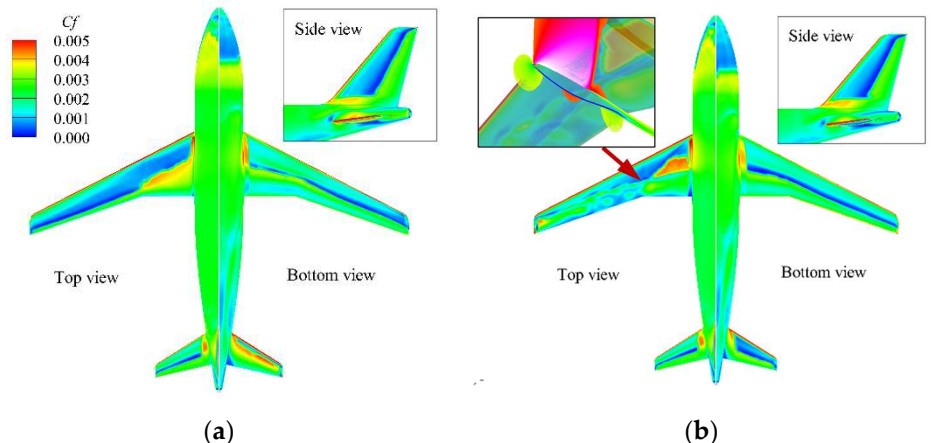

(**a**)      (**b**)

**Figure 4.** Distribution of surface friction coefficient by CFD. (**a**) $\alpha = 0°$; (**b**) $\alpha = 4°$.

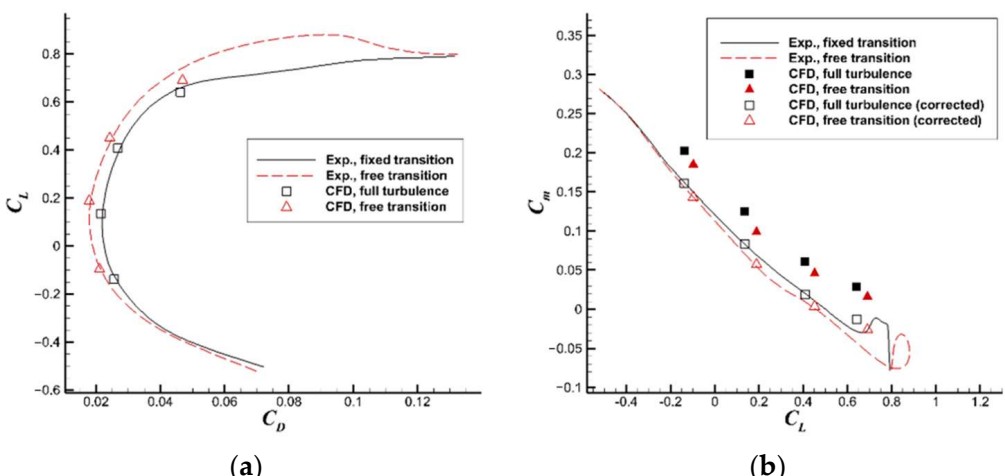

**Figure 5.** Aerodynamic characteristics by CFD and wind tunnel test. (**a**) $C_L - C_D$; (**b**) $C_m - C_L$.

From Figure 5, the calculated $C_L$ and $C_D$ are shown to be in good agreement with the test results at a small angle of attack, and $C_m$ has a difference of about 0.042. Usually, the aerodynamic interference of sting support induces a downward pitching moment that remains almost unchanged from the angle of attack before flow separation occurs [19]. Since the pitching static margin (slope of $C_m$ vs. $C_L$ curve) is the primary interest of the present study, rather than the accurate prediction of the pitching moment intercept ($C_{m0}$), the correction was made based on the difference between experimental and computational results. The corrected computational results of $C_m$ were obtained by subtracting the difference $\Delta C_m = 0.042$ from the original results. The corrected $C_m$ agrees well with the test results. There are differences between the calculated results and the test results at $\alpha = 4°$, and these are more evident when the transition location is not fixed. In Figure 4b, the serious shock/boundary layer interference on the upper surface of the wing can be seen. The near-normal shock on the leeward side of the wing causes a local separation and an abrupt thickening of the boundary layer. Therefore, an abrupt change in $C_f$ can be noted. The numerical method that was used in the present study cannot accurately simulate the flow structure of the wind tunnel test.

In general, both the full turbulence simulation based on the SST model and the free transition results based on the transition model are in good agreement with the fixed transition and free transition results of the wind tunnel test, respectively. The numerical method and model parameters can accurately predict the flow behavior and transition location of the model.

### 4.2. Influence of Temperature Gradient on Aerodynamic Characteristics

The influences of model surface temperature gradient on the lift and drag coefficients of the model were studied through Case 2. The calculation parameters are shown in Table 4. The free transition state was simulated in this case.

**Table 4.** Parameters of Case 2.

| $M$ | $Re$ | $\alpha$ | $T_{inf}$ | $T_w/T_{inf}$ |
|---|---|---|---|---|
| 0.78 | $5.0 \times 10^7$ | $0°$ | 98.1K | 0.49~3.55 |

Figure 6 shows the changes in the $C_L$ and $C_D$ of the entire model at different wall temperature ratios $T_w/T_{inf}$. $C_L$ decreases linearly with the increasing $T_w/T_{inf}$, while the drag coefficient decreases nonlinearly, faster with a lower $T_w/T_{inf}$ and slower with a higher $T_w/T_{inf}$. We assume that during an aerodynamic test in a cryogenic wind tunnel, the ratio $T_w/T_{inf}$ is about 3.0 at first and then slowly descends to 1.0. Correspondingly, the lift coefficient $C_L$ changes from 0.113 to 0.188, and the drag coefficient $C_D$ increases from

0.01378 to 0.01647. Therefore, the test results can be obviously different before and after the thermal equilibrium is reached. The relative difference between $C_L$ and $C_D$ is about 40% and 20%, respectively. Such a huge difference shows that it is not feasible to carry out the measurement when a certain degree of temperature gradient exists between the model surface and the air flow. The detailed characteristics and physical mechanisms need further investigation.

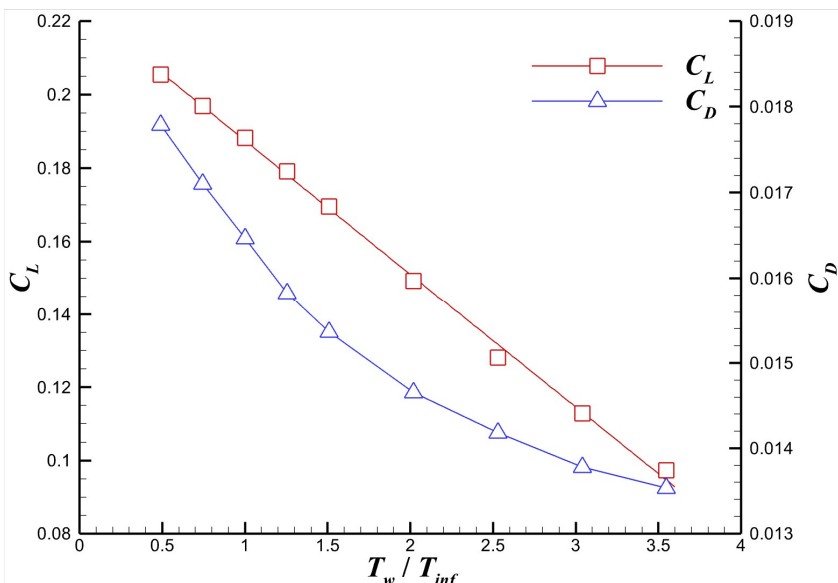

**Figure 6.** Effect of wall-temperature gradient on on $C_L$ and $C_D$ of CHN-T1.

Figure 7 provides the temperature gradient's influence on different components. As each component accounts for a different proportion of the aerodynamic coefficients of the entire model, Figure 8 shows the variation curve of the lift and drag coefficients per unit wetted area with the wall-temperature ratio. It can be seen from the figures that, for each component, the aerodynamic coefficients change with the wall-temperature ratio with the same regularity as the entire model, linearly for $C_L$ and exponentially for $C_D$.

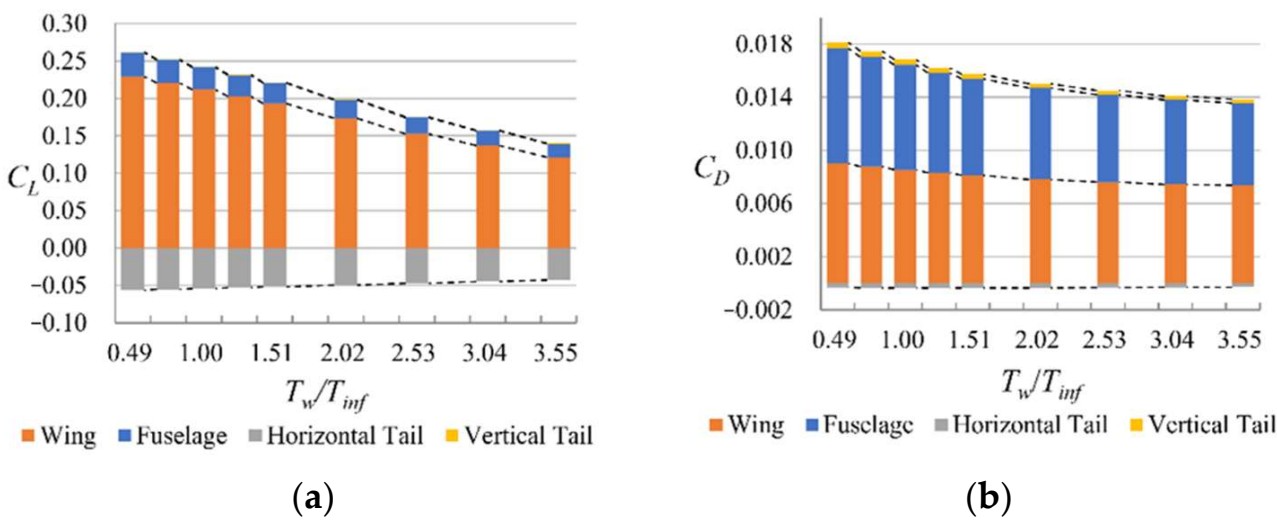

(**a**)                                                    (**b**)

**Figure 7.** Effect of wall-temperature gradient on $C_L$ and $C_D$ of each component. (**a**) $C_L - T_w/T_{inf}$; (**b**) $C_D - T_w/T_{inf}$.

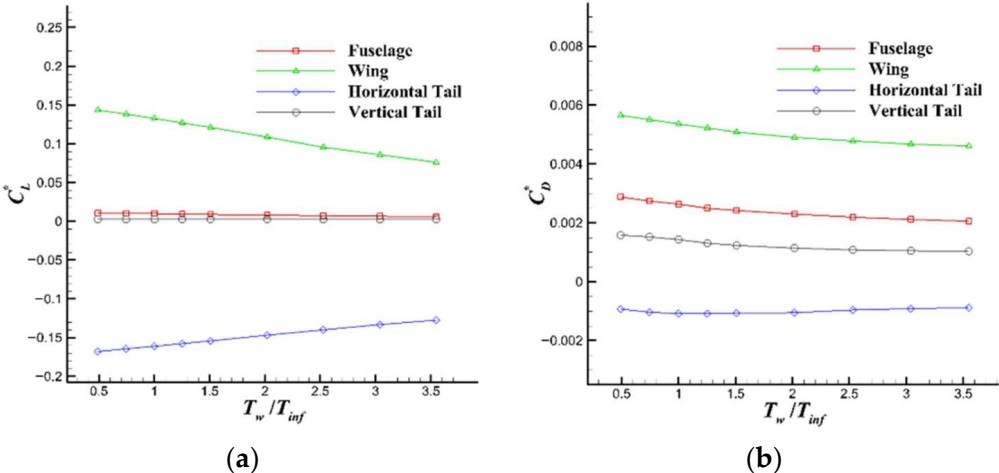

**Figure 8.** Effect of wall-temperature gradient on $C_L{}^*$ and $C_D{}^*$ of each component. (**a**) $C_L{}^* - T_w/T_{inf}$; (**b**) $C_D{}^* - T_w/T_{inf}$.

The temperature dependences of some physical properties should be discussed prior to further analysis. The skin friction coefficient $C_f$ is defined as:

$$C_f = \tau_w/0.5\rho V^2 \tag{22}$$

where the wall shear stress $\tau_w$ is determined by the viscosity and velocity gradient of the wall:

$$\tau_w = \mu(\partial u/\partial y)_w \tag{23}$$

The change in density $\rho$ with temperature $T$ follows the equation of state:

$$\rho = p/RT \tag{24}$$

where the constant $R = 287.053$ for ideal gas. Therefore, the local density decreases with an increase in temperature, causing the thickening of the boundary layer. The tangential velocity gradient near the wall then becomes smaller. However, the temperature dependence of the viscosity follows the Sutherland law:

$$\mu = \mu_0 \left(\frac{T}{T_0}\right)^{\frac{3}{2}} \left(\frac{T_0 + S}{T + S}\right) \tag{25}$$

where the reference viscosity $\mu_0 = 1.716 \times 10^{-5}$ Pa·s under the reference temperature $T_0 = 273.15$K and the Sutherland constant $S = 111.4$K. Therefore, the viscosity $\mu$ increases with increasing temperature.

The total drag can be divided into skin-friction drag and pressure drag, based on the formation mechanisms. The variation curves of the total drag, the skin-friction drag and the pressure drag of each component with the wall-temperature ratio are given in Figure 9. The influence regularity of wall-temperature gradient on different parts of the total drag is the opposite. For pressure drag, a higher wall temperature decreases the density near the wall and increases the thickness of the boundary layer, increasing the drag caused by pressure differences. For the skin-friction drag, although a higher wall temperature slightly increases the viscosity coefficient, it reduces the velocity gradient at the wall to a greater extent for the turbulence boundary layer, which leads to larger wall shear stress; therefore, the overall skin-friction drag decreases. For the wing and fuselage, the change in pressure drag is small, so the influence of temperature gradient on the skin-friction drag is the dominant factor. For the horizonal and vertical tails, the changes in pressure drag and skin-friction drag are close, resulting in a slight change in total drag. In additionas the contribution of horizonal and vertical tails to the total drag of the entire aircraft is small,

the influence that wall-temperature gradient has on the skin-friction drag of the wing and the fuselage is more important.

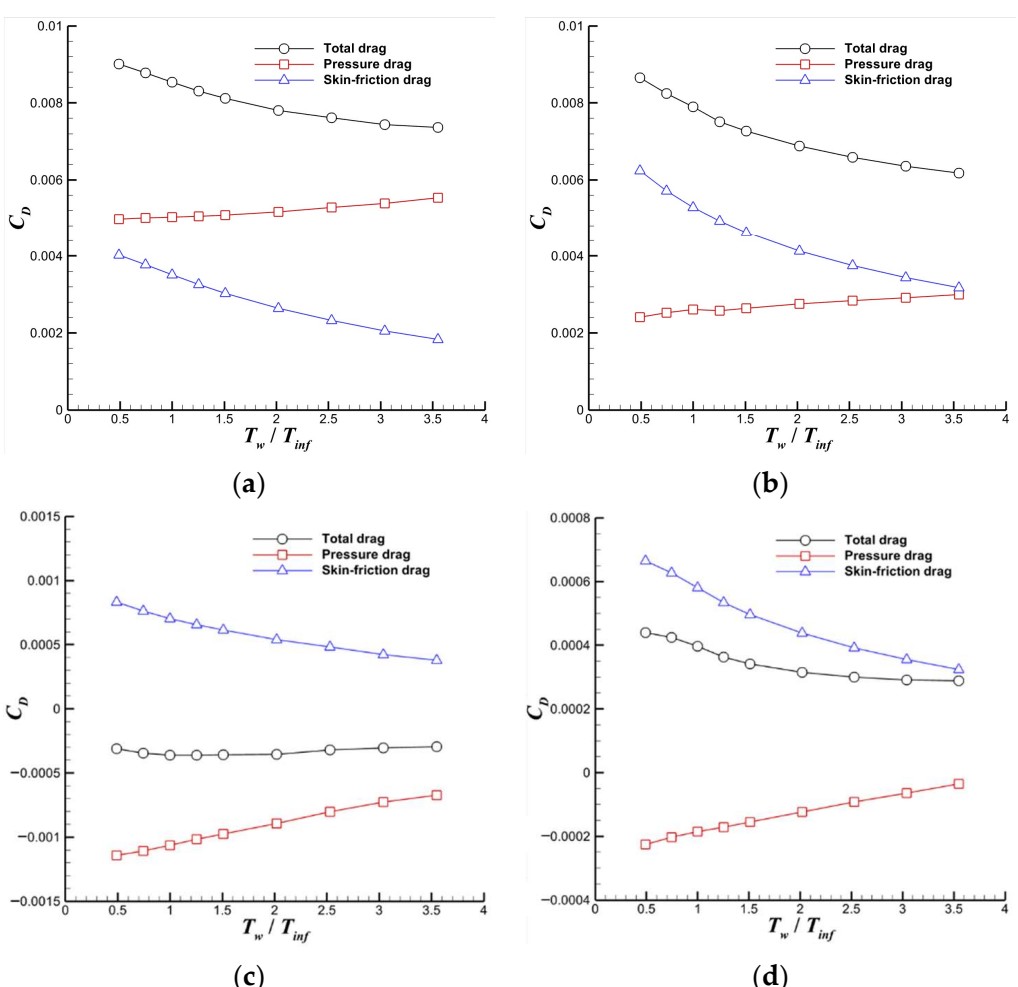

**Figure 9.** Effect of wall-temperature gradient on pressure drag and skin-friction drag. (**a**) Wing; (**b**) fuselage; (**c**) horizontal tail; (**d**) vertical tail.

### 4.3. Influence of Temperature Gradient on Flow Characteristics of Model Surface

The lift generated by the wing and the drag generated by the wing and fuselage play leading roles in the entire model, and the skin-friction drag similarly changes with the wall temperature. Therefore, the variation law of the flow characteristics around the wing was studied. According to the different spanwise sections of the wing shown in Figure 10, the distribution of the pressure coefficient is shown in Figure 11. On the upper surface of the wing, which is relatively flat, the pressure increases with the increase in wall temperature to the same extent in different chord positions. On the lower surface of the wing with relatively large curvature, the pressure decreases gradually when the wall temperature increases. When there is a large pressure gradient in the flow direction (whether favorable or adverse), the influence of wall temperature is small. The reason for the change in $C_L$ is that the displacement thickness of the boundary layer around different wing sections is affected by the varying wall temperatures and pressure gradients, so the airfoil profile changes, leading to the change in the circulation of the wing.

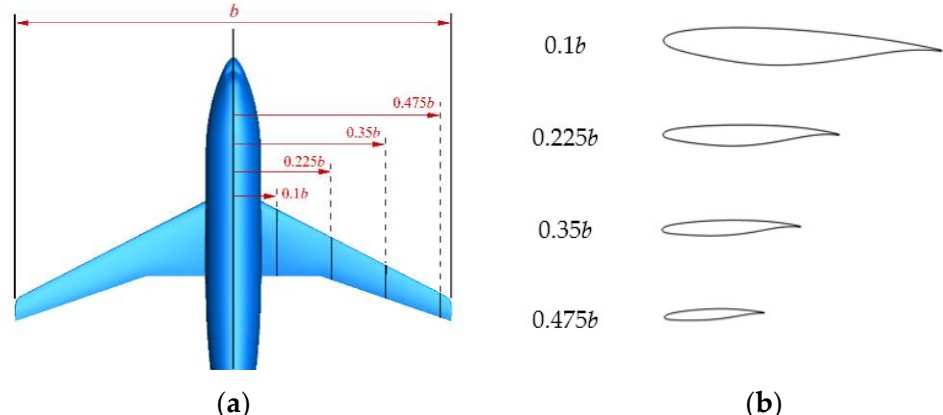

**Figure 10.** Spanwise locations and airfoil profiles of different sections on the wing. (**a**) Spanwise location; (**b**) section profiles.

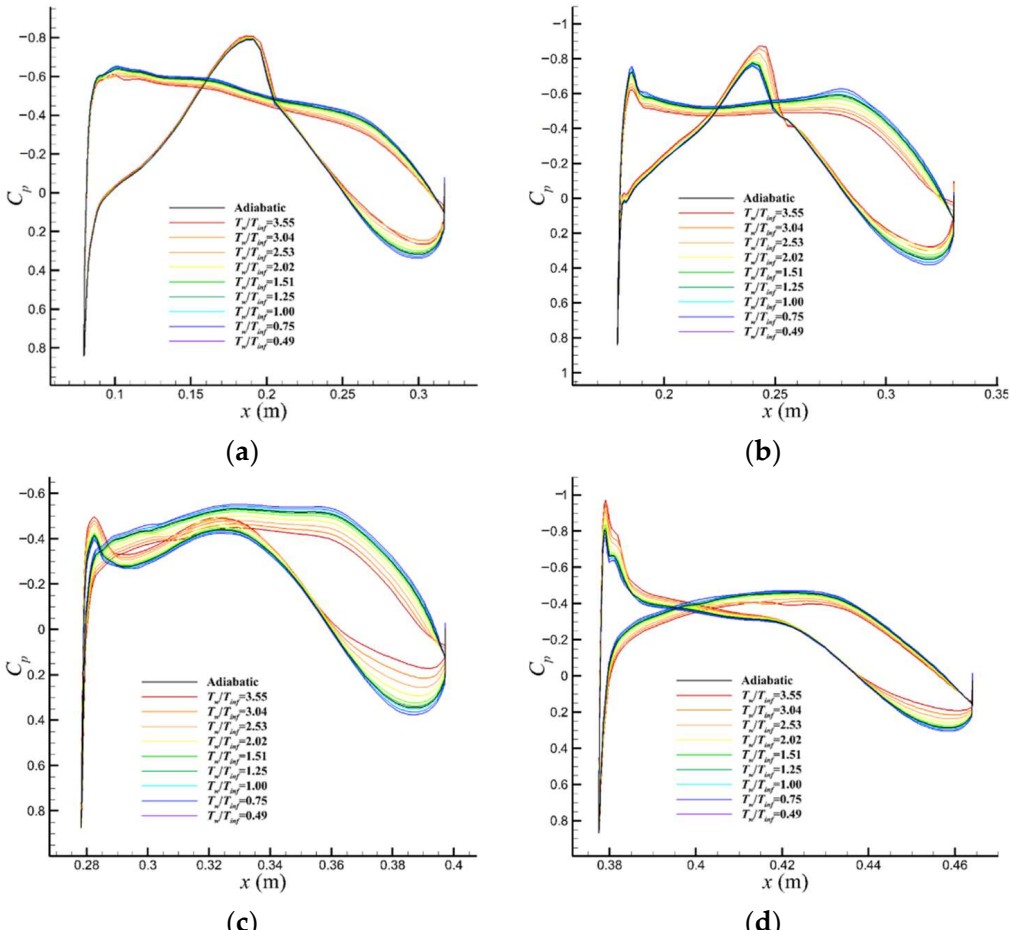

**Figure 11.** Effect of temperature gradient on pressure distributions on the wing. (**a**) Slice at 0.1b; (**b**) slice at 0.225b; (**c**) slice at 0.35b; (**d**) slice at 0.475b.

The influence of temperature gradient on the total drag of the entire model is mainly reflected by the transition position of the boundary layer and the skin-friction drag of the turbulent boundary layer. Figure 12 shows the distribution of the skin-friction coefficient on the surface under different wall-temperature conditions. When the wall is adiabatic, the laminar flow area at the leading edge of wing, horizontal tail and vertical tail is the smallest. When the wall temperature $T_w$ is equal to the incoming flow temperate $T_{inf}$, there is only a small temperature gradient between the wall and the incoming flow, so the surface flow

characteristics is almost the same as the adiabatic wall. With the gradual increase in the temperature gradient (whether the wall temperature is higher or lower), the transition position will move downstream at various degrees, increasing the laminar flow area. The transition behaviors of the inner side of the lower surface and the outer side of the upper surface of the wing are more sensitive to the change in temperature gradient. It is worth noting that when $T_w = 298.1$K, the surface flow is quite different from the adiabatic wall, and the local transition positions on the wing and the horizonal tail reach about 50% of the local chord length.

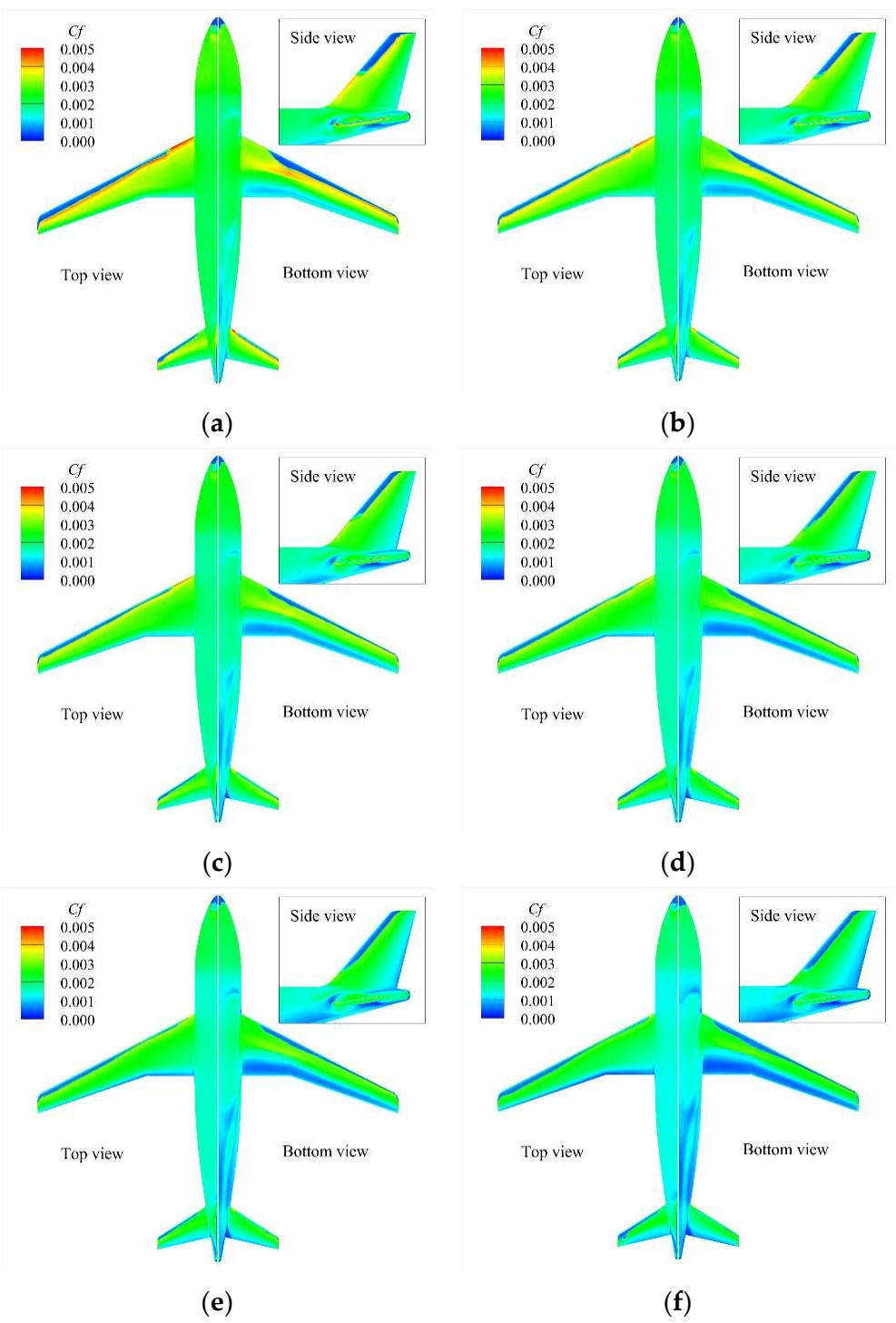

**Figure 12.** *Cont.*

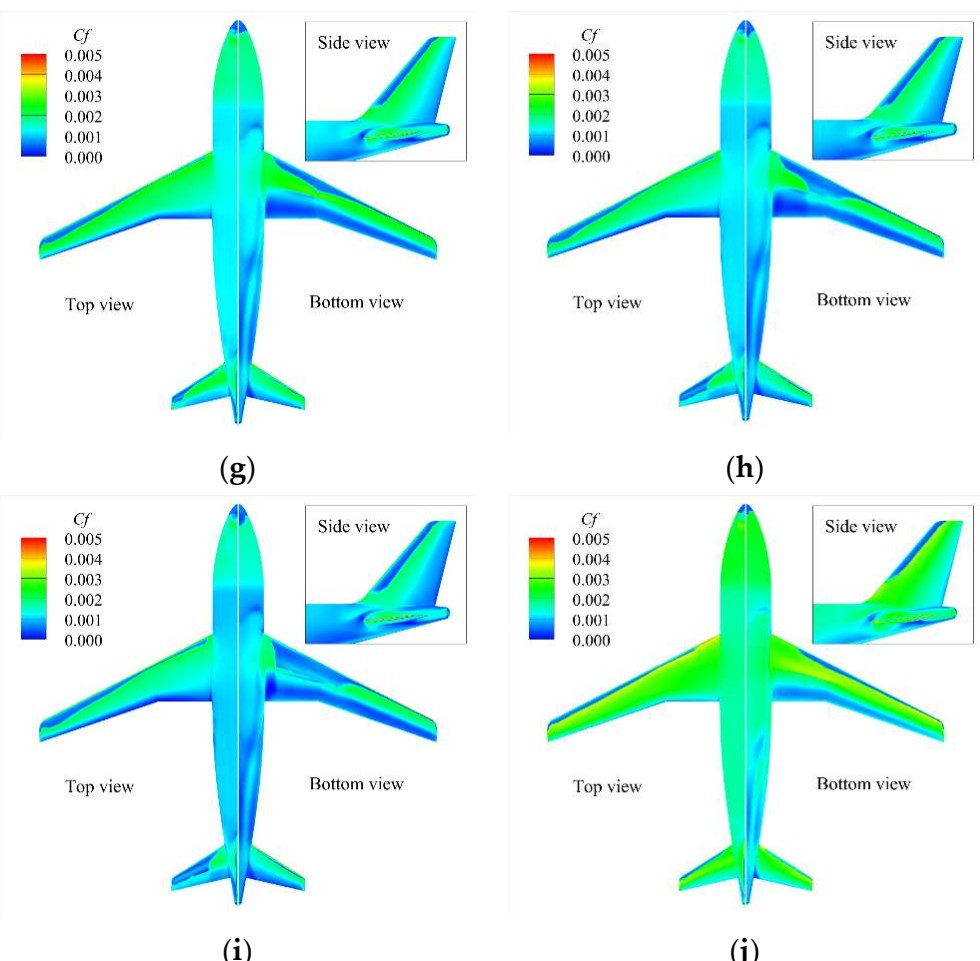

**Figure 12.** Effect of temperature gradient on skin-friction distribution. (**a**) $T_w/T_{inf}$ = 0.49; (**b**) $T_w/T_{inf}$ = 0.75; (**c**) $T_w/T_{inf}$ = 1; (**d**) $T_w/T_{inf}$ = 1.25; (**e**) $T_w/T_{inf}$ = 1.51; (**f**) $T_w/T_{inf}$ = 2.02; (**g**) $T_w/T_{inf}$ = 2.53; (**h**) $T_w/T_{inf}$ = 3.04; (**i**) $T_w/T_{inf}$ = 3.55; (**j**) adiabatic walls.

The curves of the change in transition positions on different sections of the wing with the temperature gradient are drawn in Figure 13. Due to the curvature change in the airfoil and the influence of three-dimensional flow, the curves fluctuate slightly, but a certain law can be obtained. Affected by the local angles of attack of different wing sections, the temperature gradients also have different effects on the transition position. For sections near the wing root, the transition position of the upper surface changes slightly with the increase in wall temperature, while that of the lower surface obviously moves downstream. The sections near the wing tip receive the opposite effect from wall temperature. The transition position of the upper surface moves downstream with the increase in temperature gradient, while the transition position of the lower surface changes little.

The temperature gradient affects not only the transition behavior of the laminar boundary layer, but also the friction coefficient at the wall, especially in the turbulent region. With the increase in wall temperature, the viscosity becomes larger, according to Equation (25), which increases the skin-friction coefficient, while the boundary layer becomes thicker and the velocity gradient near the wall decreases becomes of the smaller density, according to Equation (24), which reduces the skin-friction coefficient. Figure 14 shows the distributions of the skin-friction coefficient at different spanwise sections of the wing. A reverse phenomenon of temperature dependence for $C_f$ in laminar and turbulent flows can be seen because of different dominant factors. For the turbulent boundary layer, since the local thickness and the tangential velocity gradient are much larger than that of the laminar boundary layer, the change in local density is dominant, so the $C_f$ of turbulent

boundary layer decreases significantly with the increasing $T_w/T_{inf}$, while for the laminar boundary layer, the change in the local velocity gradient becomes weaker and the change in the viscosity of $\mu$ is not negligible anymore. Therefore the $C_f$ of the laminar boundary shows a slight increase when $T_w/T_{inf}$ grows.

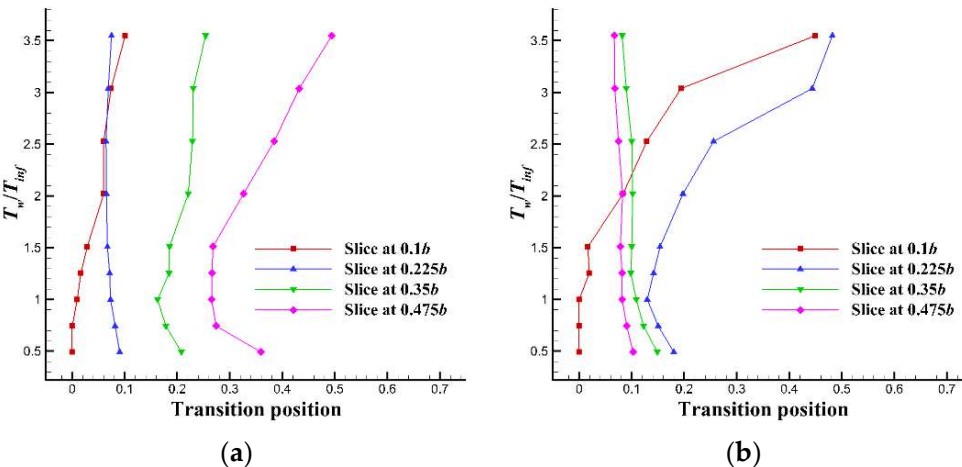

**Figure 13.** Effect of temperature gradient on transition locations. (**a**) Upper surface; (**b**) lower surface.

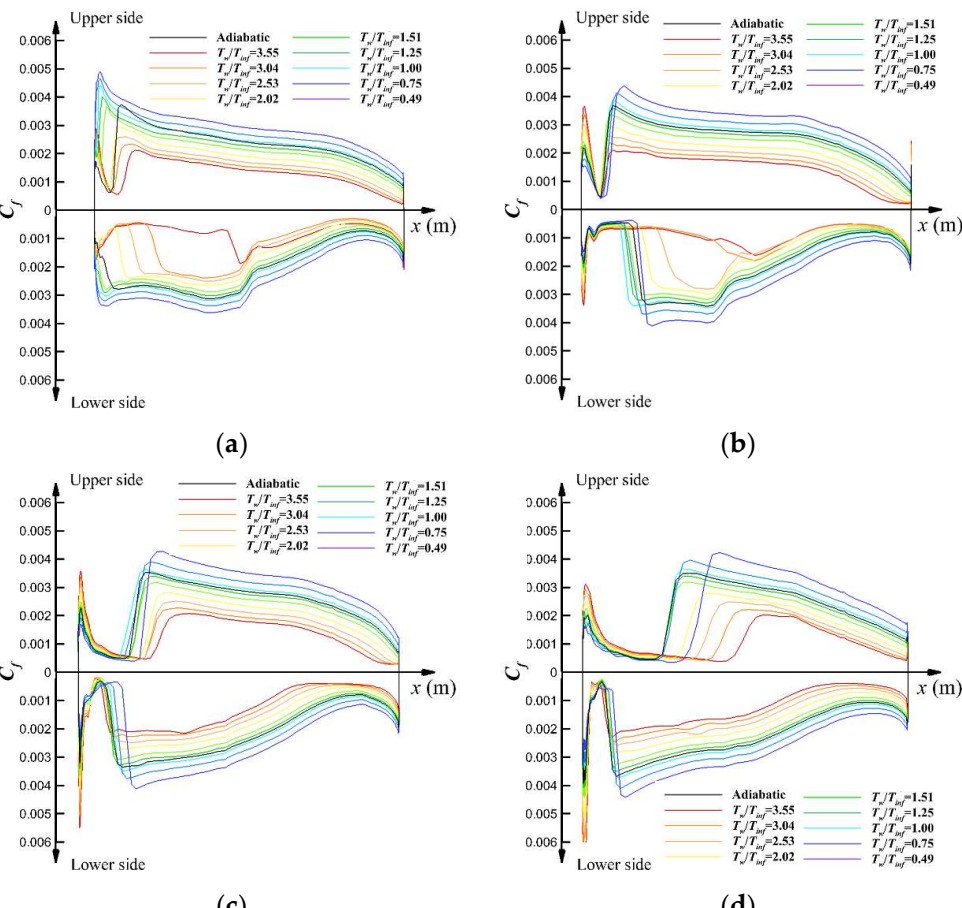

**Figure 14.** Skin-friction of different slices of the wing. (**a**) Slice at 0.1b; (**b**) slice at 0.225b; (**c**) slice at 0.35b; (**d**) slice at 0.475b.

The curve of the adiabatic wall is the closest to those found under the condition of $T_w/T_{inf} = 1$, and there is only a small difference caused by the temperature recovery near the wall.

### 4.4. Influence of Temperature Gradient on Boundary Layer Characteristics

To analyze the influence of wall-temperature gradient on the flow characteristics of the boundary layer, two points were chosen at the section $y = 0.258b$, as shown in Figure 15. For the whole range of wall temperature that the present study simulated, the two points were always located in the laminar and turbulent regions, respectively.

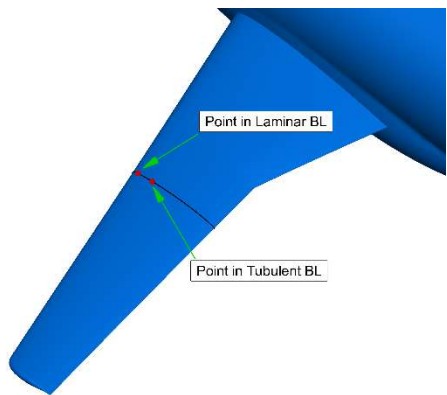

**Figure 15.** Point locations on the wing.

Figure 16 shows the influence of wall-temperature gradient on the velocity distribution of the laminar boundary layer. With the increase in wall temperature, the thickness of the boundary layer increases. In Figure 16b, $y$ is normalized by the local displacement thickness of the boundary layer. Even without considering the change in the thickness of the boundary layer, the velocity profile also becomes relatively "not full" with the increase in the wall temperature. Together with the effect of the boundary layer thickness, the velocity gradient at the wall evidently decreases in the laminar boundary layer. Figure 17 shows the influence of wall-temperature gradient on the velocity distribution of the turbulent boundary layer. The wall-temperature gradient also has an obvious effect on the velocity profile of the viscous sub-layer in the turbulent boundary layer. The velocity gradient at the wall gradually increases with the increase in the wall-temperature gradient, which is consistent with the laminar boundary layer.

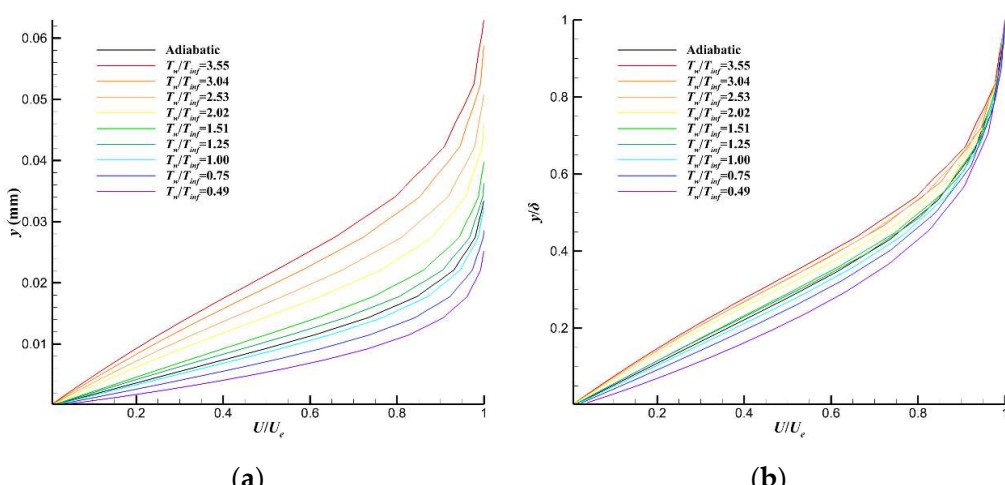

**Figure 16.** Velocity profiles of laminar boundary layers. (**a**) *U/Ue* v.s. *y*; (**b**) *U/Ue* v.s. *y/δ*.

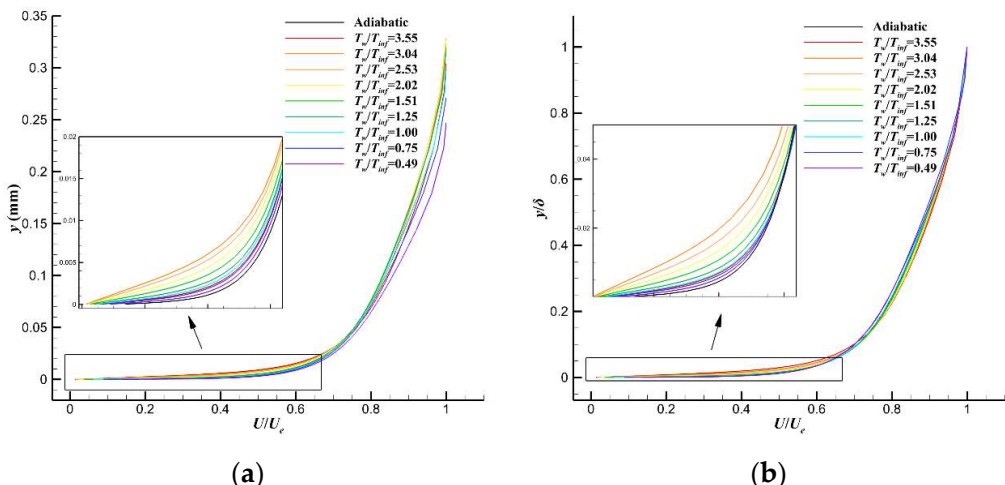

**Figure 17.** Velocity profiles of turbulent boundary layers. (**a**) *U/Ue* v.s. *y*; (**b**) *U/Ue* v.s. *y/δ*.

Figure 18 shows the effect of temperature gradient on the displacement thickness of the boundary layer at the points indicated in Figure 15. The solid lines with various symbols represent the variation in displacement thickness with the change in wall temperature in laminar and turbulent boundary layers, respectively, while the dotted lines represent the displacement thickness of the adiabatic wall. For the laminar boundary layer, the displacement thickness of the boundary layer increases linearly with the increase in wall-temperature ratio, which is consistent with the variation law of velocity type in Figure 16a. For the turbulent boundary layer, when the wall-temperature ratio is low, the displacement thickness increases quickly. However, when the wall-temperature ratio increases to a certain extent, the growth speed slows down. When $T_w/T_{inf} \geq 2.5$, the displacement thickness begins to decrease. The reason for this is that, for the turbulent boundary layer, the displacement thickness is affected not only by density changes caused by different wall temperatures, but also by the local Reynolds number of the turbulent boundary layer, which differs due to the different transition positions. From Figure 13, the transition position of the boundary layer moves rapidly downstream with the increase in wall-temperature ratio, especially when $T_w/T_{inf} \geq 2.5$, resulting in a faster reduction in the development length of the turbulent boundary layer at the selected point. Thus, it can be observed from a fixed point that the displacement thickness of the turbulent boundary layer first increases rapidly, then slows down, and finally begins to decrease with the increase in the wall-temperature ratio.

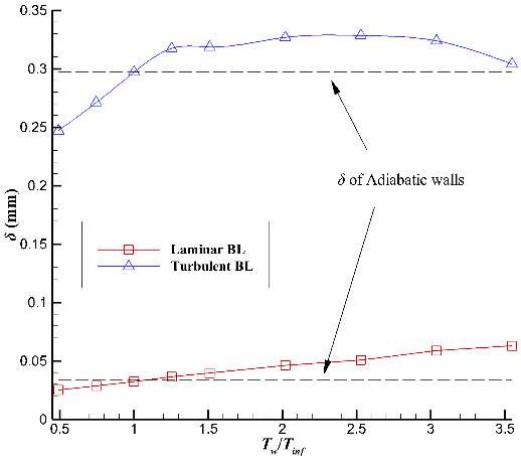

**Figure 18.** The displacement thickness of boundary layer.

Figures 19 and 20 show the influence of wall temperature on the temperature distribution of laminar and turbulent boundary layers, respectively. The relative change in the displacement thickness of the laminar boundary layer is larger, so the influence of the nondimensionalization of the normal distance $y$ of the boundary layer can be clearly seen from Figure 18. For the adiabatic wall, the temperature gradient in the normal direction at the wall is zero, and the temperature at the wall is different from that of the incoming flow. The relation can be described as follows [20]:

$$T_{w,aw} = T_e \left( 1 + r\frac{\gamma - 1}{2} M_e^2 \right) \tag{26}$$

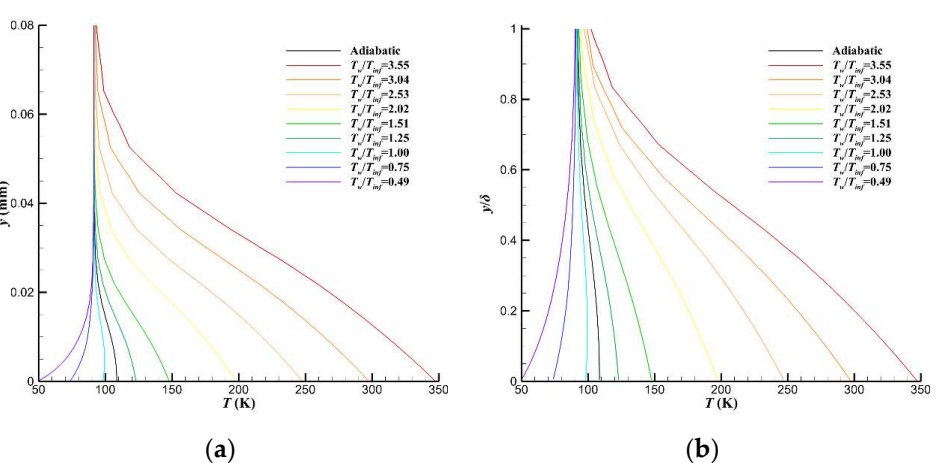

(a)   (b)

**Figure 19.** Temperature distribution of laminar boundary layers. (**a**) $T$ v.s. $y$; (**b**) $T$ v.s. $y/\delta$.

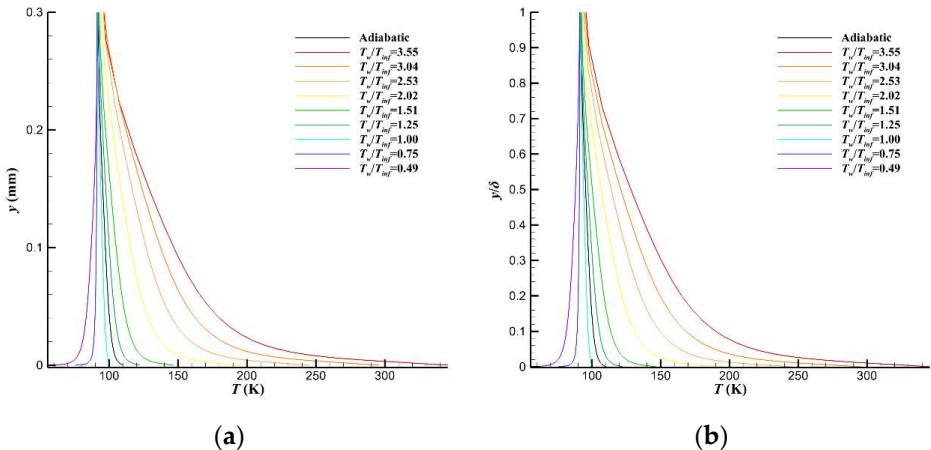

(a)   (b)

**Figure 20.** Temperature distributions of turbulent boundary layers. (**a**) $T$ v.s. $y$; (**b**) $T$ v.s. $y/\delta$.

The recovery factors $r$ for laminar and turbulent boundary layers are $r_{lam} = 0.85$ and $r_{tur} = 0.90$. In the present case, $M_e \approx M_{inf} = 0.78$, $T_e \approx T_{inf} = 98.1$K; thus, the wall temperature under adiabatic conditions for laminar and turbulent boundary layers can be predicted as $T_{w,aw,lam} \approx 108.2$K and $T_{w,aw,tur} \approx 108.8$K, which are highly consistent with the results shown by the black curves in Figures 19 and 20.

When the wall temperature is higher than the adiabatic wall temperature, the gradient at the wall is negative, and the temperature decreases in the normal direction to the incoming flow temperature. When the wall temperature is lower than the adiabatic temperature, the gradient is positive, and the temperature increases to the incoming flow. For the turbulent boundary layer, most of the temperature changes are compressed in the viscous sub-layer, so the change in temperature gradient at the wall is inconspicuous.

Figure 21 shows the distribution of turbulent kinetic energy $k$ in the turbulent boundary layer. In the viscous sub-layer, when the wall temperature is higher, because of energy injections from the wall, the turbulent kinetic energy can reach a higher peak, while the growth of turbulent kinetic energy in the normal direction becomes slower as the viscous sub-layer is relatively thicker. In the buffer layer and the logarithmic-law layer, the higher peak in turbulent kinetic energy means that it will begin to dissipate at a faster speed. Another key influence factor on the dissipating speed is the displacement thickness. A larger temperature gradient thickens the local boundary layer and slows down the dissipation. Therefore, the dissipation process first becomes slower, then becomes faster with the increase in wall-temperature gradient.

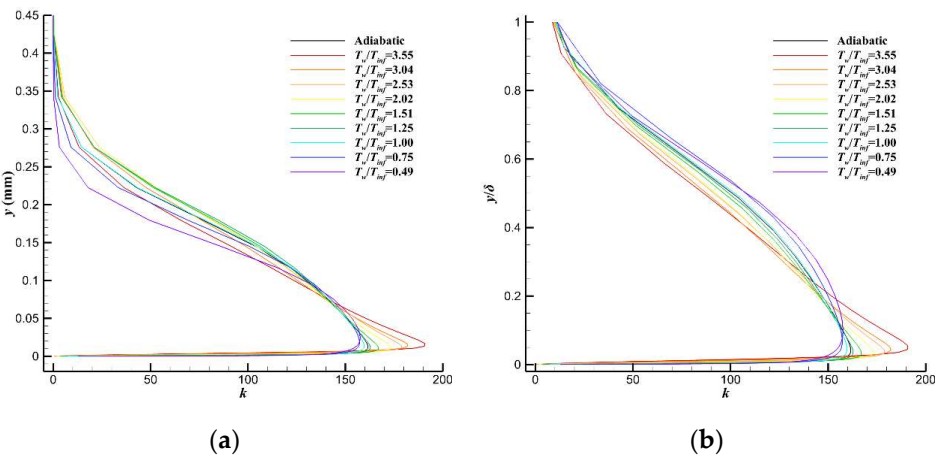

(a)  (b)

**Figure 21.** Turbulent kinetic energy distribution of different temperature gradient. (**a**) $k$ v.s. $y$; (**b**) $k$ v.s. $y/\delta$.

## 5. Conclusions

A numerical simulation of the CHN-T1 standard model of different wall-temperature ratios was conducted using Langtry–Menter $\gamma$-$Re_{\theta t}$ transition model to investigate the nonadiabatic wall effect on the static aerodynamic coefficients in a cryogenic wind tunnel. The wall temperature's effects on lift and drag coefficients, together with the surface and boundary layer flow characteristics, were investigated. The main conclusions are as follows:

(a) The change in wall-temperature gradient has a significant impact on the lift and drag coefficients of the entire model. With the increase in wall-temperature ratio $T_w/T_{inf}$, the lift and drag coefficients gradually decrease with different variation laws.

(b) The wall-temperature gradient has a certain impact on the pressure distribution on the upper and lower surfaces of the wing. With the increase in wall-temperature ratio $T_w/T_{inf}$, the upper surface pressure increases and the lower surface pressure decreases.

(c) The wall-temperature gradient affects the transition position of the model surface. For particular spanwise locations of the wing, with the increase in wall-temperature ratio $T_w/T_{inf}$, the transition position moves downstream.

(d) The skin-friction coefficient decreases with the increase in wall temperature as the velocity profile and displacement thickness of the boundary layer are evidently affected by the nonzero temperature gradient at the wall.

**Author Contributions:** Conceptualization, N.X.; Methodology, X.W., J.W., G.L. and Y.T.; Validation, X.W., Z.Z. and Y.T.; Formal analysis, J.C.; Investigation, X.W. and Y.T.; Resources, J.C. and G.L.; Data curation, J.W., Y.L. and N.X.; Writing—original draft preparation, Y.T., J.W., J.C. and Y.L.; Writing—review & editing, X.W., Z.Z., G.L. and N.X. All authors have read and agreed to the published version of the manuscript.

**Funding:** This research received no external funding.

**Data Availability Statement:** The data presented in this study are available on request from the corresponding author.

**Conflicts of Interest:** The authors declare no conflict of interest.

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
