# Peer review of "Numerical Study of Nonadiabatic Wall Effects on Aerodynamic Characteristics of CHN-T1 Standard Model"

_aerospace, doi:10.3390/aerospace10040372_

Round 1

Reviewer 1 Report

The authors have investigated the effects of differences in temperature between the incoming ambient air and the aircraft wall surface on compressible flow, boundary layer, and flight characteristics, while using turbulence and transition models for a standard aircraft model. Such investigations provide the necessary knowledge to correctly evaluate the location of turbulent transitions and flight characteristics at the aircraft surface in wind tunnel experiments conducted at room temperature, for example, and to make reasonable predictions about conditions during high-altitude flight. In fact, this paper demonstrates that the transition position changes predominantly in response to the difference (or ratio) with the airframe surface temperature, and also discusses the factors that contribute to this change in the boundary layer thickness. In fact, this paper demonstrates that the transition position changes predominantly in response to the difference (or ratio) with the airframe surface temperature, and also discusses the factors that contribute to this change in the boundary layer thickness.

However, prior to publication, the authors should be requested to consider the following additional explanations or modifications:

- The temperature dependences of the physical properties including density and viscosity are the key factors of this study. Therefore, those dependences or correlations with the temperature should be clearly described in Section 2 with mathematical formulas.

- All specific values of model constants and expressions for model functions the authors used should be clearly stated in the manuscript. Or, appropriate references should be cited. As you may know, often, the CFD results will vary depending on the turbulence model.

- As most readers can guess, isn’t it necessary to include the defining equations and names for CL, Cd, and Cm, firstly shown in Table 1?

- At line 187, the authors said “When the difference is corrected, …”, but the details of how to correct it are not mentioned. Please clarify.

- At line 189-191, “From Figure 3(b), the serious shock/boundary layer interference … can be seen”. Please highlight the relevant position in the figure.

- Regarding Figure 8(c,d), is the principle of the negative shape drag force generation obvious? I am not familiar with such a negative drag.

- As can be seen in Figure 13, the temperature dependence of Cf is reversed for laminar and turbulent flows. This point should be noted and discussed in the discussion. Sometimes the authors discussed without distinguishing between laminar and turbulent flows, which may lead to misunderstanding.

- Is the measurement location for Figure 17 same with those fir Figures 15 and 16? This point is unclear.

- I could not understand the final sentence of “forming a more complex change regularity” at line 369.

- Line 394-418: Acknowledgements, etc. should be appropriate and template sentences should be removed.

Reviewer 2 Report

The manuscript presents the study of the effect of non-adiabatic wall effects on the aerodynamic characteristics of the CHN-T1 test model in a cyrogenic wind tunnel. The paper is thorough and well-written, with important conclusions having implications in the field of wind tunnel experimentation. I recommend this paper for publication. 

Author Response

Thanks for your thorough review and professional comments. Some revisions have been made to improving this paper. Please see the revised version.

Reviewer 3 Report

General comments: This paper presents important information regarding the wall temperature gradient effects.. The conducted work is quite good and gives good co-relation with exp results... However, the paper in it's current form suffers from major deficits in quality of presentation and written English... The methodologies are not well described.. Several discussions are not clear due to the poor descriptions. In many graphs, the independent variable is confused with other variables.. Visualizations presented need to be improved and further visualizations such as section cut-plane contours or vectors  through the wing for pressure distribution, velocity etc will be informative.. The presented results are of good scientific strength, but again needs improvement in discussions and style of presentation.. The conclusions are generally well written, but some quantification is needed to make a specific conclusion. 

The paper could also use some more references especially in the first three to four paragraphs in the introduction.

0) What is the Novelty of the paper? Has this analysis been not conducted in CFD before? 

1) What was the CFD code that was used? 

2) Section 2 , Computational scheme should be renamed to possibly Computational framework, followed by at least one paragraph before subsections

3) L110 - Venkatakrishnan*

4) L113 - Lower-Upper Symmetric Gauss-Seidel (LU-SGS). *

5) Section 3.2 - Some comments on the wall y+ approach adopted is necessary,  since this is a computational paper, more information on the number of cells along trailing edge (blunt or sharp?) on each grid level is necessary, what was the %local spacing (based on chord or span) used at fuselage/wing intersection and along span (min, max).

6) Section 3.2 - There are no plots/graphs of convergence of forces or residuals... 

7) Section 3.2 - There is no indication of the time step adopted for the simulations, what kind of time integration approach is used? Has a time step independence study been conducted?

8) In essence, a more detailed computational framework is necessary to be given. 

9) Fig 3 - the range chosen for skin friction coefficient is questionable. Perhaps a better range would be from 0.0005 to 0.005. Right now the red colour is completely invisible.

10) Figure 3 label should be improved to mention skin friction coefficient, C_f, ****

11) Fig 3 - upward view should be rephrased to something more clear for any reader** do you mean bottom view?

12)  Fig 3 - I am guessing the authors made this grid in ICEM CFD (I could be wrong), but for your paper improvement it would be nice to see a scan plan chordwise for your chosen grid level.. perhaps include as another figure or as a subfigure.

13)  Fig 4 and it's discussions, no clear explanations of how the data was corrected, although good co-relation with WT results.

14) Line 203-Line 215 , discussions of Fig 5 is poor, the plot independent variable is Tw/Tinf, and therefore make the discussions using this variable, the paragraph in it's current form is not clear... Also careful where you use room temperature, wall temperature and Tinf.

15) Do you have a reference for Eqn 9 in Line 351?

16) Fig 15 and Fig 16, the graph caption mentions U/Ue vs y and U/Ue vs y/delta... but the plots are made as y vs U/Ue and y/delta vs U/Ue? What is the correct independent variable here? This variable should be on x-axis..

Round 2

Reviewer 1 Report

Now, I can recommend its publication.